# Impacts of Dental Follicle Cells and Periodontal Ligament Cells on the Bone Invasion of Well-Differentiated Oral Squamous Cell Carcinoma

**DOI:** 10.3390/cancers17091559

**Published:** 2025-05-03

**Authors:** Anqi Chang, Kiyofumi Takabatake, Tianyan Piao, Takuma Arashima, Hotaka Kawai, Htoo Shwe Eain, Yamin Soe, Zin Zin Min, Keisuke Nakano, Hitoshi Nagatsuka

**Affiliations:** Department of Oral Pathology and Medicine, Okayama University, 2-5-1 Shikata-cho, Kita-ku, Okayama 700-8525, Japan; pmff7em4@s.okayama-u.ac.jp (A.C.); pfd12zwy@s.okayama-u.ac.jp (T.P.); de428001@s.okayama-u.ac.jp (T.A.); de18018@s.okayama-u.ac.jp (H.K.); pmp61kpp@s.okayama-u.ac.jp (H.S.E.); pki31ld5@s.okayama-u.ac.jp (Y.S.); pobk1heq@s.okayama-u.ac.jp (Z.Z.M.); pir19btp@okayama-u.ac.jp (K.N.); jin@okayama-u.ac.jp (H.N.)

**Keywords:** oral squamous cell carcinoma, dental follicle cells, periodontal ligament cells, bone invasion, receptor activator of NF-κB ligand, parathyroid hormone-related peptide

## Abstract

Oral squamous cell carcinoma is a common head and neck cancer that often spreads into the jawbone, leading to severe complications. Understanding the factors that promote or inhibit this process is essential for improving treatment strategies. In this study, we examined two types of dental cells—dental follicle cells and periodontal ligament cells—to explore their effects on cancer-associated bone invasion. Our results show that dental follicle cells enhance bone destruction by stimulating bone-resorbing cells (osteoclasts), whereas periodontal ligament cells suppress this process, potentially protecting against tumor invasion. These findings reveal how stromal cells influence cancer progression and suggest that targeting these interactions could lead to new therapeutic approaches for bone-invasive oral cancer.

## 1. Introduction

Oral squamous cell carcinoma (OSCC) is the most prevalent malignancy in the head and neck region, commonly affecting the lips, buccal mucosa, tongue, and floor of the mouth. Among its subtypes, gingival oral squamous cell carcinoma (GOSCC) is the second most frequent, following tongue carcinoma [1,2,3]. OSCC, particularly GOSCC, tends to invade both soft and hard tissues, including gingival connective tissue and alveolar bone. The anatomical origin of the tumor significantly influences its invasive behavior and clinical prognosis [2,3]. Bone invasion is more frequently observed in tumors adjacent to bony structures or in advanced stages. For instance, gingival carcinomas demonstrate a higher tendency to invade the jawbone compared to lingual carcinomas. Clinically, bone invasion is associated with poorer prognosis and presents considerable challenges in diagnosis and treatment planning [4].

The progression of bone-invasive OSCC is primarily mediated by osteoclast activation, which is largely regulated by the receptor activator of the NF-κB ligand (RANKL). Fibroblastic cells at the tumor invasion front have been shown to express RANKL and interleukin-6 (IL-6), both of which stimulate osteoclast differentiation and function, leading to progressive bone destruction [5,6,7]. Previous studies, including our own, have demonstrated that stromal cell components play a critical role in OSCC bone invasion by modulating tumor invasiveness, epithelial–mesenchymal transition (EMT), and the expression of RANKL and parathyroid hormone-related peptide (PTHrP) [8,9]. However, the specific contributions of mesenchymal-derived stromal cells—particularly dental follicle cells (DFCs) and periodontal ligament cells (PDLCs)—to OSCC-induced bone resorption have not been systematically studied. Given their physiological roles in bone remodeling, elucidating how DFCs and PDLCs modulate cancer-induced osteolysis represents an important yet underexplored area of OSCC research.

DFCs and PDLCs are both mesenchymal-derived cell populations with established roles in bone remodeling and periodontal tissue regeneration. DFCs comprise a heterogeneous population of fibroblasts and stem cells that can differentiate into cementoblasts, periodontal ligament cells, and alveolar bone-forming osteoblasts. They are essential for tooth eruption and have been shown to promote osteoclast activation during this process, facilitating alveolar bone remodeling [10,11,12,13]. The osteogenic potential of DFCs is regulated by signaling pathways such as Wnt/β-catenin, with bone morphogenetic proteins (BMP-2, BMP-6, and BMP-9) enhancing their differentiation into osteoblasts, whereas inflammatory cytokines like IL-1α inhibit osteogenesis and promote osteoclast activity [14,15,16,17,18]. Despite their physiological relevance, the potential involvement of DFCs in OSCC-associated bone destruction has not been well characterized in the context of cancer-induced bone resorption, especially in OSCC, leaving an important gap in current knowledge.

PDLCs, located in the periodontal ligament, are another population of multipotent mesenchymal cells capable of differentiating into adipocytes, fibroblasts, cementoblasts, and osteoblasts [19,20,21,22]. When transplanted into immunodeficient mice, PDLCs can regenerate structures resembling periodontal ligament and cementum, supporting their potential in periodontal tissue engineering [23,24,25]. Compared with other mesenchymal stem cell-derived populations, PDLCs exhibit morphology and functional traits that closely resemble native periodontal tissue, indicating their importance in maintaining periodontal homeostasis. In our previous study using an OSCC xenograft model, PDLCs were found to inhibit osteoclast activation and reduce tumor-induced bone destruction, suggesting a potential suppressive role of PDLCs in the context of cancer-associated bone resorption [9]. However, the underlying mechanisms by which PDLCs modulate OSCC-induced bone invasion remain largely unexplored and warrant further investigation.

Given the distinct roles of DFCs and PDLCs in bone remodeling and periodontal regeneration, we hypothesized that these cells may influence OSCC bone invasion through their interactions with tumor cells and the surrounding microenvironment. To investigate this, we isolated DFCs and PDLCs from the dental follicles of impacted third molars and evaluated their effects on OSCC-associated bone destruction using in vitro co-culture systems and an in vivo xenograft mouse model. Additionally, to establish a comparative reference for osteogenic capacity, we included the KUSA cell line—a well-characterized pre-osteoblast cell line with robust osteogenic potential—as a positive control in mineral deposition assays and ALP activity assays. By elucidating the regulatory roles of these stromal cells, our study aims to advance understanding of the mechanisms underlying OSCC-associated bone resorption and identify potential stromal targets for therapeutic intervention in bone-invasive OSCC. To our knowledge, this is among the first studies to systematically compare the distinct effects of DFCs and PDLCs on OSCC-induced bone resorption.

## 2. Materials and Methods

### 2.1. Isolation and Culture of Dental Follicle Cells and Periodontal Ligament Cells

Primary DFCs and PDLCs were isolated from two healthy, non-smoking adult male patients who underwent third molar extraction at the Department of Oral and Maxillofacial Surgery, Okayama University. Both donors were confirmed to be free of systemic disease, long-term medication use, and oral pathological conditions, as determined by radiographic and clinical examination. Following informed consent and based on the protocol of our previous studies, the samples were collected under sterile conditions. Specifically, DFCs were harvested from the dental follicle tissues surrounding the crown and cervical regions of completely impacted, unerupted third molars, as these structures are only present prior to eruption. In contrast, PDLCs were isolated from the periodontal ligament tissue along the root surface of partially or fully erupted third molars [8,9,26]. Tissues were cut into ~1 mm³ pieces and washed several times with α-MEM (Life Technologies, Thermo Fisher Scientific Inc., KK, Tokyo, Japan) and 1% antimycotic-antibiotic (Life Technologies, Thermo Fisher Scientific Inc., KK, Tokyo, Japan). These fragments were enzymatically digested in α-MEM containing 1 mg/mL collagenase II and dispase (both from Invitrogen, Thermo Fisher Scientific Inc., KK, Tokyo, Japan) for 2 h at 37 °C with continuous agitation (200 rpm). The dissociated cells were filtered through a 100 µm cell strainer (Falcon, Corning Life Sciences, Corning, NY, USA), centrifuged at 111.8× *g* for 5 min, and resuspended in α-MEM supplemented with 10% fetal bovine serum (FBS, Biowest, Nuaillé, France). After incubation at 37 °C in 5% CO_2_ for one week, adherent epithelial and stromal cells were separated using Accutase (Invitrogen; Thermo Fisher Scientific Inc., Waltham, MA, USA) based on their differential adhesion properties, as described previously [8,9,26]. All cells were used within 10 passages. The cells were all maintained at 37 °C in a humidified environment of 95% air and 5% CO_2_. This research was approved by the Ethics Committee of Okayama University (Project ID: 1703-042-001, approval granted on 10 March 2017). Additionally, all the human tissue was procured from the patients with written informed consent.

### 2.2. OSCC Cell Line, KUSA Cell Line and Macrophage Cell Line

The well-differentiated human OSCC cell line HSC-2 (JCRB, Osaka, Japan) and the murine macrophage cell line RAW264.7 (RCB0535) were obtained from the Japanese Collection of Research Bioresources (JCRB, Osaka, Japan) and the RIKEN Bioresource Center Cell Bank (Tsukuba, Ibaraki, Japan), respectively. KUSA, a murine pre-osteoblastic cell line with fibrocyte-like morphology, was established from bone marrow stromal cells derived from primary femoral bone cultures of female C3H/He mice. This cell line was kindly provided by Dr. Akihiro Umezawa of Keio University (Tokyo, Japan).

### 2.3. Giemsa Staining

When cell cultures reached ~90% confluency, DFCs and PDLCs were seeded in the 6-well plates containing one coverslip each well (22 mm × 22 mm; Matsunami Glass Ind., Ltd., Osaka, Japan) at a density of 3 × 10^5^ cells per well and incubated at 37 °C. After 2 days, the slides attached with cells were gathered and then stained using a Giemsa staining kit (Diff-Quick; Sysmex Corporation, Kobe, Japan) following the manufacturer’s protocol. Images were captured using a bright-field microscope (BX51; Olympus Corporation, Tokyo, Japan) at 40× magnification.

### 2.4. Immunofluorescence (IF) Staining

After the cell density reached approximately 90%, DFCs and PDLCs were plated on 6-well plates, including one coverslip per well (22 mm × 22 mm; Matsunami Glass Ind., Ltd., Osaka, Japan) at the density of 3 × 10^5^ cells per well. Two days later, the slides were fixed with 4% paraformaldehyde (PFA; Nacalai Tesque, Kyoto, Japan) for 15 min, blocked with blocking buffer (DS Pharma Biomedical Co., Ltd., Osaka, Japan) for 20 min, and washed three times with TBS. Afterwards, the primary antibodies: Vimentin anti-rabbit (ab16700, SP20, 1:200, Abcam, Cambridge, UK; a mesenchymal marker), AE1/3 anti-mouse (IS053, AE1/3, Abcam, Cambridge, UK; an epithelial marker), CD73 anti-rabbit (Abcam, Cambridge, UK; a marker of MSCs), CD90 anti-rat (Abcam, Cambridge, UK; a marker of MSCs) and CD105 anti-mouse (DAKO, Glostrup, Denmark; a marker of MSCs) were used to incubate with these slides for 1 h followed by rinsing triple times by TBS. Subsequently, the slides were further incubated with secondary antibodies: anti-mouse IgG Alexa Fluor 488 (A21441, 1:200, Thermo Fisher Scientific Inc., Waltham, MA, USA), anti-rabbit IgG Alexa Fluor 568 (A10042, 1:200, Thermo Fisher Scientific Inc., Waltham, MA, USA) and anti-rat IgG Alexa Fluor 488 (A21441, 1:200, Thermo Fisher Scientific Inc., Waltham, MA, USA) for 1 h at room temperature in a lucifugal circumstance. After that, all the slides were stained with 0.2 g/mL 40,6-diamidino-2-phe-nylindole (DAPI; Dojindo Molecular Technologies, Inc., Kumamoto, Japan), followed by triple washes in TBS and distilled water (DW). Images were captured using a fluorescence microscope (BZ-8000; Keyence Corp., Osaka, Japan) at 5× and 10× magnifications.

### 2.5. Mineralization Assay by Alizarin Red

To evaluate the osteogenic differentiation potential, DFCs, PDLCs, and KUSA were seeded at a density of 1 × 10⁵ cells per well in 12-well plates and cultured in α-MEM supplemented with 10% fetal bovine serum (FBS; Biowest, Nuaillé, France), totally 2 mL. When the cells reached approximately 90% confluency, the culture medium was replaced with 2 mL of osteogenic induction medium consisting of α-MEM, 10% FBS, 10 mM β-glycerophosphate (Sigma-Aldrich, St. Louis, MO, USA), and 0.05 mM L-ascorbic acid (Sigma-Aldrich, St. Louis, MO, USA). After incubation for 3 weeks, cells were rinsed with PBS and fixed in 95% ethanol at 37 °C for 15 min, then stained with 1% alizarin red S (Katayama Chemical Industries Co., Ltd., Osaka, Japan) for 15 min. KUSA cells exhibit mature osteoblast-like properties and are capable of consistent osteogenic differentiation both in vitro and in vivo [27]. Due to their robust and reproducible mineralization capacity, KUSA cells are widely used as a positive control in osteogenic assays. In this study, they were included as a positive control to benchmark the mineral deposition of DFCs and PDLCs.

### 2.6. Alkaline Phosphatase (ALP) Assay

Upon reaching confluency, DFCs, PDLCs, and KUSA were switched to osteogenic medium. ALP activity was measured on day 5 using the p-Nitrophenyl Phosphatase Substrate method (FUJIFILM Wako Pure Chemical Co., Osaka, Japan) following the manufacturer’s protocol. KUSA cells were used as a positive control in this assay due to their robust and reproducible osteogenic potential, particularly their high ALP activity during the early stages of differentiation [27]. Their inclusion allowed us to benchmark the osteogenic response of DFCs and PDLCs under identical conditions.

### 2.7. Tartrate-Resistant Acid Phosphatase (TRAP) Staining for Cells

When cell cultures reached ~90% confluency, RAW264.7, HSC-2, and DFCs/PDLCs were harvested separately using ethylenediaminetetraacetic acid (EDTA; Thermo Fisher Scientific, Inc., Waltham, MA, USA) or Accutase (Innovative Cell Technologies, Inc., San Diego, CA, USA). The cells were mixed at a ratio of 3:3:1 (RAW264.7:DFCs/PDLCs:HSC-2) and plated in 6-well plates with coverslips (22 mm × 22 mm; Matsunami Glass Ind., Ltd., Osaka, Japan) at a density of 3.0 × 10⁵ cells per well. After 3 days of incubation, slides attached with cells were conducted with TRAP staining by TRAP staining kit (cat. no. AK04F; Cosmo Bio Co., Ltd., Tokyo, Japan) [28,29]. Finally, with the aim of analyzing the positive-osteoclast (TRAP-positive osteoclasts) percentage and the cell growth of osteoclasts, we randomly captured ten images (40× magnification, BX51; Olympus Corporation, Tokyo, Japan) of each slide and calculated the numbers by ImageJ software (version 1.53K; National Institutes of Health, Bethesda, MD, USA). The independent experiments were repeated in triplicate. This seeding ratio was also adjusted according to the time each cell type takes to reach confluence in monoculture to ensure synchronized growth in co-culture conditions.

### 2.8. Construction of Animal Model

This study was conducted in strict accordance with the ethical guidelines approved by the Institutional Animal Care and Use Committee of Okayama University (Approval No. OKU-2022354). To ensure animal welfare, predefined humane endpoints were established, including impaired eating or drinking, signs of distress (such as self-harm, abnormal posture, or respiratory issues), persistent physical abnormalities without signs of recovery (e.g., diarrhea, bleeding, or vulvar soiling), rapid weight loss exceeding 20% over several days, or tumor size reaching ≥1 cm in diameter. The experiment was immediately terminated, and humane euthanasia was performed if any signs of intolerable pain were observed.

Anesthesia was administered using isoflurane inhalation in accordance with Okayama University Animal Experiment Committee guidelines. Induction was performed at 5% isoflurane, and sedation was maintained at 2–3%. Adequate anesthesia was confirmed by assessing whether mice could regain a prone position when placed in a supine posture. All procedures were approved by the Ethics Committee. Following anesthesia, a total of 200 μL cell suspension (HSC-2, 1 × 10⁶, 100 μL; DFCs/PDLCs, 3 × 10⁶, 100 μL) was carefully injected into the subcutaneous tissue over the skull of 15 healthy 7-week-old female BALB/c nu-nu mice (15 g; Shimizu Laboratory Supplies Co., Ltd., Kyoto, Japan). A sample size of five animals per group (*n* = 5) was chosen based on standard practices in previous studies and ethical considerations. This number balances statistical power and biological reproducibility with the need to minimize animal use in accordance with the 3Rs principle (Replacement, Reduction, and Refinement). It was sufficient to detect significant differences across experimental conditions while adhering to institutional animal care guidelines. All experiments were conducted in an accredited facility under standard environmental conditions: 25 °C, 50–60% humidity, and a 12 h light/dark cycle. Mice had free access to food and water and were randomly assigned to three groups (*n* = 5 per group): HSC-2 only, HSC-2+DFCs, and HSC-2+PDLCs.

### 2.9. Hematoxylin and Eosin (HE) Staining

After four weeks, all mice were humanely euthanized by isoflurane overdose (>5%), and cardiac arrest was confirmed via pulse palpation before performing cervical dislocation. Tumor tissues were excised, fixed in 4% paraformaldehyde (PFA; Nacalai Tesque, Kyoto, Japan) for 12 h at room temperature, and then decalcified in 10% ethylenediaminetetraacetic acid (EDTA; Thermo Fisher Scientific, Inc., Waltham, MA, USA) at 4 °C for 4 weeks. Tissue samples were paraffin-embedded, sectioned into 5-μm slices, and conducted with HE staining. Images were obtained using a bright-field microscope (BX51, Olympus Corporation, Tokyo, Japan) at 40× magnification.

### 2.10. TRAP Staining for Tissues

The 5-μm sections were further used for TRAP staining according to the manufacturer’s instructions using the aforementioned TRAP staining kit (cat. no. AK04F; Cosmo Bio Co., Ltd., Tokyo, Japan). The stained sections were photographed with a bright-field microscope (40× magnification; BX51; Olympus Corporation, Tokyo, Japan), and the TRAP-positive cells were counted using ImageJ software (version 1.53 K; National Institutes of Health, Bethesda, MD, USA).

### 2.11. Immunohistochemistry (IHC) Staining

Followed by antigen retrieval with 0.01 M trisodium citrate buffer (pH 6; FUJIFILM Wako Pure Chemical Corporation, Osaka, Japan) for 1 min in a microwave, 5 μm-thick sections were blocked with 10% normal serum (DS Pharma Biomedical Co., Ltd., Osaka, Japan) for 20 min at room temperature. The sections were then incubated overnight at 4 °C with primary antibodies: RANKL anti-rabbit (cat. no. bs-0747R; 1:100; Bioss Antibodies, Beijing, China) and PTHrP anti-rabbit (cat. no. 10817-1-AP; 1:100; Proteintech Group Inc., Rosemont, IL, USA). Following three washes with TBS, the sections were incubated for 1 h at room temperature with the secondary antibody (PK-6101, rabbit ABC kit; Vector Laboratories, Newark, CA, USA) and subsequently visualized using a diaminobenzidine (DAB)/H_2_O_2_ solution (Histofine DAB substrate; Nichirei Biosciences Inc., Tokyo, Japan). To evaluate RANKL and PTHrP expression, five images (40× magnification) were captured per sample, and IHC scores were also calculated. The IHC score was determined by multiplying the positive-cell percentage score by the staining-intensity score. The positive cell percentage was scored as follows: 0 (<1%), 1 (1–24%), 2 (25–49%), 3 (50–74%), and 4 (75–100%). Staining intensity was scored as follows: 0 (no staining), 1 (weak, light yellow), 2 (moderate, brown), and 3 (strong, dark brown) [8,9].

### 2.12. Statistics Analysis

Statistical analyses were performed using GraphPad Prism 9 software (GraphPad Software, San Diego, CA, USA). Cell-based experiments were conducted in biological triplicate, while animal experiments were repeated three times per group. This group size was selected based on the previous literature and ethical considerations, ensuring a balance between statistical power and the principles of the 3Rs (Replacement, Reduction, and Refinement) in animal research. To determine the appropriate statistical method, the Shapiro–Wilk test was first performed to assess the normality of data distribution. As some datasets did not follow a normal distribution, non-parametric tests were chosen. Specifically, the Kruskal–Wallis test was used for overall comparisons, followed by the Steel–Dwass post hoc test for multiple pairwise analyses. All data were presented as mean ± SD, and a *p*-value < 0.05 was considered statistically significant.

## 3. Results

### 3.1. Characteristics of DFCs and PDLCs

Figure 1a presents the morphological characteristics and marker expression profiles of DFCs and PDLCs, demonstrating similar spindle-shaped morphology in Giemsa staining. To confirm their cellular composition, double immunofluorescence staining for AE1/3 and Vimentin was performed. Both cell types were positive for Vimentin but negative for AE1/3, indicating that they were predominantly mesenchymal in origin, with no detectable epithelial contamination.

Since DFCs and PDLCs have been reported to contain stem cell populations, we further evaluated their biological characteristics using double immunofluorescence staining for CD73 and CD105 and CD73 and CD90. The results showed that both DFCs and PDLCs expressed CD73 but lacked expression of CD90 and CD105.

Alizarin red staining was then performed to assess the mineral matrix deposition ability of DFCs and PDLCs following three weeks of osteogenic induction. Among all experimental groups, KUSA cells exhibited the highest mineral deposition, whereas no significant differences were observed between DFCs and PDLCs (Figure 1b).

Furthermore, ALP activity was measured after five days of culture in an osteogenic medium to assess early osteogenic differentiation. The results showed that KUSA cells displayed significantly higher ALP activity compared to both DFCs and PDLCs, with no notable differences between DFCs and PDLCs (Figure 1c).

### 3.2. Effects of the Interaction Between DFCs, PDLCs, and HSC-2 on Proliferation, Differentiation, and Osteoclast Activation In Vitro and In Vivo

The impact of DFCs and PDLCs on RAW264.7 cells was assessed to evaluate their roles in osteoclast proliferation and differentiation. As a control experiment, the effects of DFCs and PDLCs on RAW264.7 cells without interaction with HSC-2 were examined in vitro using TRAP staining (Figure 2a). Among the three groups, some TRAP-positive cells with mononuclear morphology were observed. The total number of RAW264.7 cells per image was higher in the RAW264.7+DFCs group compared to both the RAW264.7 only group and the RAW264.7+PDLCs group (Figure 2b). Interestingly, the RAW264.7+DFCs group exhibited the lowest percentage of TRAP-positive RAW264.7 cells, suggesting that DFCs predominantly promoted RAW264.7 proliferation rather than osteoclast differentiation (Figure 2c). In contrast, PDLCs facilitated the differentiation of RAW264.7 cells into osteoclasts.

Notably, the RAW264.7+PDLCs group showed a markedly reduced total cell count despite being cultured under the same conditions and initial seeding density. This may reflect the inhibitory effect of PDLCs on RAW264.7 proliferation. Moreover, as terminal osteoclast differentiation is generally associated with reduced proliferative capacity, the increased differentiation observed in PDLC-containing groups may inherently result in fewer RAW264.7 cells overall [30].

To further assess the effects of HSC-2 interaction, TRAP staining was performed on RAW264.7 cells co-cultured with DFCs or PDLCs in the presence of HSC-2 (Figure 2d). In all three groups, large, multinucleated TRAP-positive cells were observed. Compared to the other groups, the RAW264.7+DFCs+HSC-2 group exhibited both the highest total RAW264.7 cell count and the highest percentage of TRAP-positive RAW264.7 cells, indicating that DFCs, in combination with HSC-2, enhanced both RAW264.7 proliferation and differentiation into osteoclasts (Figure 2e,f). Conversely, the RAW264.7+PDLCs+HSC-2 group showed suppressed proliferation and differentiation of RAW264.7 cells into osteoclasts, further supporting the inhibitory role of PDLCs in osteoclastogenesis.

In addition to the in vitro findings, in vivo experiments were conducted. HE staining was performed to evaluate tumor differentiation and bone surface resorption (Figure 3a). The severity of bone resorption followed the order: HSC-2+PDLCs < HSC-2 only < HSC-2+DFCs, indicating that DFCs promoted more extensive bone degradation than PDLCs. To further confirm osteoclast activation, TRAP staining was applied to analyze osteoclast activation in vivo (Figure 3b). The HSC-2+DFCs group displayed a greater number and larger size of TRAP-positive cells compared to the HSC-2-only group, whereas the HSC-2+PDLCs group showed fewer and smaller TRAP-positive cells than the other two groups. Quantitative analysis confirmed that the HSC-2+DFCs group had the highest number of TRAP-positive cells attached to the bone surface (Figure 3c).

Collectively, these findings indicate that the interactions between HSC-2 and DFCs facilitate osteoclast differentiation both in vitro and in vivo, whereas the interactions between HSC-2 and PDLCs inhibit osteoclast differentiation under the same conditions.

### 3.3. Effects of DFCs and PDLCs on RANKL and PTHrP Expression in HSC-2 In Vitro and In Vivo

Since osteoclast activation is primarily regulated by RANKL and PTHrP, double immunofluorescence staining for AE1/3 and RANKL and AE1/3 and PTHrP was conducted to assess RANKL and PTHrP expression in HSC-2 cells (Figure 4a and Figure 5a). Quantitative analysis revealed that both the number and percentage of RANKL(+)AE1/3(+) double-positive cells were significantly higher in the HSC-2+DFCs group compared to the HSC-2 only and HSC-2+PDLCs groups, with statistical significance (Figure 4b,c). PDLCs co-culture produced a contrasting profile: although it raised the relative percentage of PTHrP(+)AE1/3(+) cells to a level comparable with the HSC-2+DFCs condition (Figure 5c), it markedly reduced the absolute number of those cells and, most importantly, suppressed RANKL expression across every quantitative parameter (Figure 4b,c,e).

To further validate these findings in vivo, IHC staining was conducted to assess RANKL and PTHrP expression in cancer cells located at OSCC bone invasion sites (Figure 4d and Figure 5d). IHC analysis demonstrated that the IHC scores of RANKL and PTHrP were the highest in the HSC-2+DFCs group, followed by the HSC-2 only group, whereas the HSC-2+PDLCs group exhibited the lowest IHC scores (Figure 4e and Figure 5e).

Collectively, these results demonstrate that DFCs robustly enhance RANKL and PTHrP expression in OSCC cells at bone-invasion sites, thereby amplifying osteoclast-activating signals, whereas PDLCs exert the opposite effect—sharply down-regulating RANKL while limiting the overall PTHrP output, despite a relative enrichment of PTHrP-positive cancer cells.

## 4. Discussion

In this study, we demonstrated that DFCs and PDLCs, two types of tooth-associated fibroblasts, exhibited distinct effects on bone resorption in OSCC. Our findings revealed that interactions between HSC-2 and DFCs enhanced OSCC bone invasion in vivo and promoted RAW264.7 differentiation into osteoclasts in vitro. In contrast, interactions between HSC-2 and PDLCs suppressed OSCC bone invasion in vivo and inhibited RAW264.7 differentiation into osteoclasts in vitro. Furthermore, DFCs upregulated RANKL and PTHrP expression in OSCC cells at bone invasion fronts, while PDLCs exerted an opposing effect. Although both DFCs and PDLCs exhibited spindle-shaped fibroblast-like morphology and CD73-positive mesenchymal stem cell (MSC)-like features, their functional roles in OSCC-induced bone destruction were markedly different.

During tooth eruption, DFCs facilitate this process by activating osteoclasts while also contributing to alveolar bone formation via osteogenic differentiation into osteoblasts [12,13]. Recent studies have highlighted the osteogenic potential of DFCs [31,32], which is regulated by signaling molecules such as BMP-2, BMP-6, and BMP-9. Inflammatory cytokines like IL-1α inhibit DFC-mediated osteogenesis and concurrently enhance osteoclast activity [15,16,17,18]. Additionally, Runx2, a key transcription factor in DFCs, governs bone remodeling and tooth eruption. Its expression can be downregulated by microRNA-204, which suppresses osteogenic differentiation by targeting Runx2, ALP, and SPARC [33,34]. Based on these mechanisms, we hypothesized that cytokines secreted by OSCC cells, including IL-1α, may target DFCs, impairing their osteogenic differentiation while promoting osteoclast activation. Our results support this hypothesis: DFCs, through their interaction with OSCC, appear to amplify osteoclastogenesis and diminish osteogenic activity, creating a pro-resorptive tumor microenvironment that exacerbates bone destruction.

PDLCs, by contrast, are essential for periodontal homeostasis and anchoring the tooth to alveolar bone [35]. These cells have been shown to express RANKL and promote the formation of multinucleated osteoclast-like cells [35,36,37]. However, they also secrete osteoprotegerin (OPG), a decoy receptor for RANKL, which inhibits osteoclastogenesis [38,39]. While previous studies suggest that co-culturing PDLCs and DFCs can promote the differentiation of both osteoblasts and osteoclasts, our findings indicate that the isolated PDLCs used in this study reduced osteoclast activity, suggesting they did not actively promote osteoclastogenesis under the tested conditions. One possible explanation is the absence of epithelial or vascular endothelial cells, which are reported to exert protective effects against osteoclast-mediated bone resorption [30]. Notably, our PDLCs lacked epithelial components, as confirmed by AE1/3 staining (Figure 1). Still, it is possible that PDLCs interact with epithelial tumor cells (OSCC) in a manner that indirectly suppresses osteoclast activity and bone resorption. Interestingly, in our previous study using a similar xenograft model, we found that gingival stromal cells (G-SCs) promoted osteolysis, whereas periodontal ligament stromal cells (P-SCs)—which correspond to the PDLCs used in the current study—exhibited an inhibitory effect on OSCC-induced bone resorption [9]. This finding is consistent with the present results and suggests that PDLCs may play a protective role in the tumor–bone microenvironment by attenuating osteoclast activation.

Our data confirmed the osteogenic potential of both DFCs and PDLCs, as demonstrated by mineral deposition observed in Alizarin Red staining (Figure 1b) and ALP activity following osteogenic induction (Figure 1c). Although these assays qualitatively support osteogenic differentiation, we acknowledge that no quantitative image analysis (e.g., using ImageJ) was performed for Figure 1b, which may limit the objectivity of this result. Interestingly, despite their inherent osteogenic capability, both DFCs and PDLCs appeared to participate in osteoclast activation and bone resorption when exposed to OSCC cells. This paradox highlights one of the most intriguing findings of our study—that stromal cells typically involved in bone formation may also contribute to bone destruction within the tumor microenvironment.

However, it is important to note that the primary aim of this study was to investigate the osteoclastogenic influence of stromal–tumor interactions rather than to comprehensively assess the osteogenic profiles of DFCs and PDLCs. For this reason, we did not further evaluate molecular markers such as Runx2, ALP, or OCN. Future research should address this limitation by incorporating quantitative and molecular analyses to elucidate how these cells regulate the balance between bone formation and resorption in OSCC.

Quantitative immunohistochemistry revealed that co-culture with DFCs significantly increased the number, percentage, and IHC scores of RANKL(+)AE1/3(+) and PTHrP(+)AE1/3(+) cells, suggesting a broad upregulation of osteoclast-activating signals by DFCs. In contrast, PDLCs displayed a divergent pattern—while they markedly suppressed RANKL expression across all parameters, they simultaneously enhanced the relative proportion of PTHrP(+)AE1/3(+) cells (Figure 4b,c,e and Figure 5b,c,e). Notably, in the HSC-2+PDLCs group, the absolute number of PTHrP(+)AE1/3(+) cells was reduced (Figure 5b), but their proportion among the total cancer cell population increased (Figure 5c), despite a concurrent decline in overall IHC scores (Figure 5e). This paradox may reflect a global inhibitory effect of PDLCs on HSC-2 proliferation and protein synthesis, leading to a reduced tumor mass while selectively enriching PTHrP-expressing subpopulations. Such stromal cell-induced phenotypic selection has been previously reported in tumor heterogeneity mediated by TGF-β and BMP signaling [40,41].

Furthermore, immunohistochemical analysis confirmed that RANKL and PTHrP expression was predominantly localized within AE1/3-positive OSCC regions, with minimal staining observed in the surrounding stroma. These results support the conclusion that OSCC cells are the primary source of osteoclast-activating factors in this model. Prior studies have demonstrated that OSCC cells, including the HSC-2 line, can secrete RANKL and PTHrP through autocrine and paracrine pathways to stimulate osteoclastogenesis [42,43]. Our findings build upon this understanding by highlighting how tumor–stromal interactions, particularly involving DFCs and PDLCs, dynamically modulate the expression of these key mediators in the tumor microenvironment.

Moreover, although DFCs are best known for their role during tooth eruption and are more prevalent in younger individuals, recent studies confirm the presence of functional DFCs even in adult-impacted molars [12,13]. In our study, DFCs were derived from adults (ages 22 and 24) and retained osteoclast-activating potential and MSC-like features. This supports the idea that adult DFCs can respond to inflammatory or tumor-derived cues and actively modulate bone homeostasis [11,12]. Nonetheless, future studies should explore age- and stage-dependent variation in DFC function.

To visualize the mechanisms by which DFCs and PDLCs regulate OSCC-associated bone resorption, we included a schematic model (Figure 6). In this model, DFCs enhance the expression of RANKL and PTHrP in OSCC cells through tumor–stroma interactions, thereby promoting osteoclast activation and accelerating bone invasion. In contrast, PDLCs do not significantly induce the expression of these osteoclast-activating factors—particularly RANKL—and are associated with reduced osteoclast formation and bone destruction. These opposing effects suggest that DFCs may promote, while PDLCs may suppress bone resorption in the OSCC microenvironment. This schematic underscores the heterogeneity of stromal contributions and highlights the therapeutic potential of targeting specific stromal–tumor interactions in OSCC.

Despite these insights, several limitations should be acknowledged. First, mineral deposition was not quantitatively analyzed (e.g., using ImageJ), which may limit the objectivity of the osteogenic potential assessment. Second, although both DFCs and PDLCs exhibited early osteogenic markers and mineralization capacity, our subsequent experiments primarily focused on osteoclast-related mechanisms, and the differences in osteogenic marker expression (e.g., Runx2, OCN, ALP) between the two cell types were not further explored. Third, although we confirmed the functional characteristics of adult-derived DFCs, this study did not stratify DFC and PDLC activity across age groups or tumor stages. Fourth, while our immunohistochemical results suggested that RANKL and PTHrP expression was largely confined to AE1/3-positive OSCC tumor regions, their precise spatial relationship with osteoclast activation remains unclear. Future studies employing dual-marker staining or serial-section analysis will be needed to map these interactions more accurately. Lastly, our conclusions are based on simplified in vitro and subcutaneous xenograft models, which may not fully recapitulate the complex OSCC–bone microenvironment in patients. These limitations should be addressed in future investigations to further validate and expand upon our findings.

Future studies should aim to elucidate the precise molecular circuits through which DFCs and PDLCs regulate osteoclastogenesis in the OSCC microenvironment. To achieve this, a combination of cytokine profiling, single-cell transcriptomic analysis, and functional validation in age-matched in vivo models will be essential. Moreover, investigating how patient-specific factors, such as donor age or inflammatory status, influence the behavior of these stromal cells may provide further insight into their context-dependent roles. These approaches will help lay the groundwork for developing stroma-targeted therapeutic strategies to attenuate tumor-induced bone destruction and improve clinical outcomes in OSCC patients.

## 5. Conclusions

In this study, we demonstrated that DFCs and PDLCs—two tooth-derived mesenchymal stromal populations—exerted distinct and contrasting effects on bone invasion in oral squamous cell carcinoma (OSCC). DFCs significantly enhanced osteoclast activation and OSCC-induced bone resorption by upregulating key osteolytic mediators, RANKL and PTHrP. In contrast, PDLCs exhibited a selective regulatory role: while they did not suppress PTHrP expression to the same extent, they markedly downregulated RANKL expression, resulting in reduced osteoclast activity and an overall attenuation of tumor-associated bone destruction.

These findings highlight the importance of stromal heterogeneity in the OSCC microenvironment and suggest that mesenchymal stromal cells of different origins contribute uniquely to bone remodeling and tumor progression. Future studies incorporating single-cell transcriptomics, spatial mapping, and immune-stroma interaction profiling will be instrumental in further unraveling the stromal dynamics in bone-invasive OSCC and in identifying cell-type-specific therapeutic targets.

## Figures and Tables

**Figure 1 cancers-17-01559-f001:**
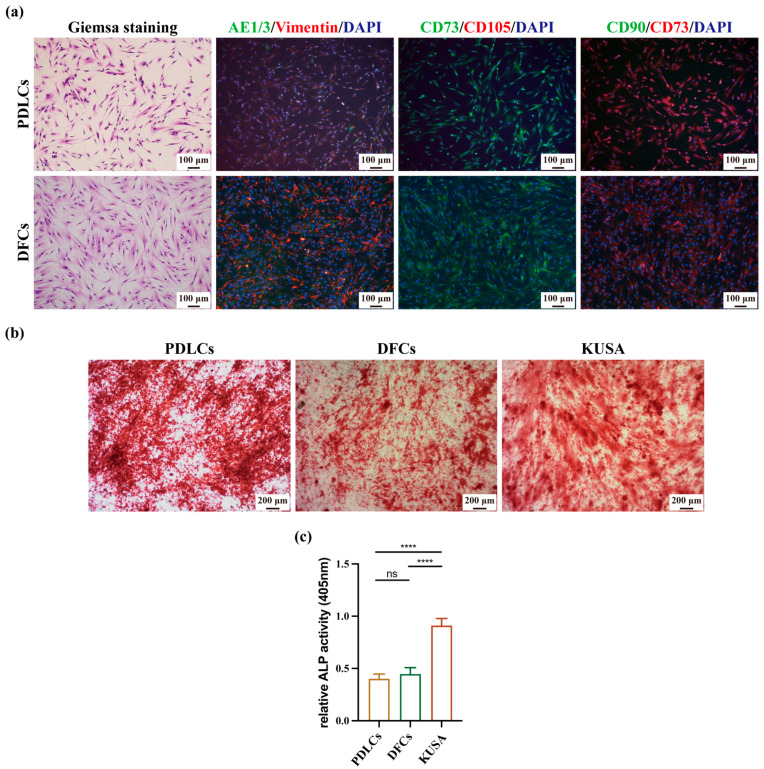
Biological characterization of DFCs and PDLCs. (**a**) Giemsa staining was performed to examine the morphological characteristics of DFCs and PDLCs. Double immunofluorescence (IF) staining for AE1/3 and Vimentin was used to confirm their cellular composition, while CD73 and CD105 and CD90 and CD73 double IF staining were conducted to assess their stem cell-like properties. As the primary aim of these immunofluorescence assays was to confirm cell identity and mesenchymal features, fluorescence intensity was not quantitatively analyzed. (**b**) Alizarin Red staining was performed after three weeks of osteogenic induction to evaluate mineral matrix deposition in DFCs and PDLCs. (**c**) Alkaline phosphatase (ALP) activity was quantified in DFCs and PDLCs cultured in osteogenic medium. **** *p* < 0.0001.

**Figure 2 cancers-17-01559-f002:**
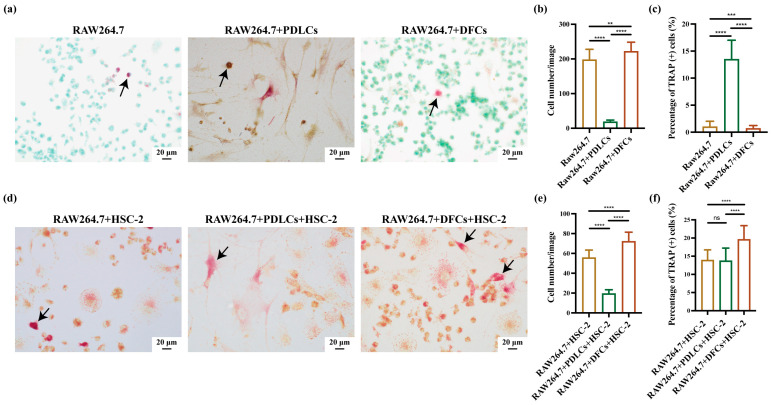
Effects of the crosstalk between DFCs/PDLCs and HSC-2 on osteoclast proliferation, differentiation, and activation in vitro. (**a**) TRAP staining was applied to evaluate TRAP-positive cells in the absence of HSC-2 interactions (RAW264.7, RAW264.7+PDLCs, and RAW264.7+DFCs groups). Black arrows indicated representative TRAP-positive osteoclasts. (**b**) Total RAW264.7 cell number and (**c**) percentage of TRAP-positive RAW264.7 cells were quantified. Data were presented as mean ± SD from three independent experiments. ** *p* < 0.01, *** *p* < 0.001, **** *p* < 0.0001. (**d**) TRAP staining was also conducted to assess TRAP-positive cells in the presence of HSC-2 interactions (RAW264.7+HSC-2, RAW264.7+PDLCs+HSC-2, and RAW264.7+DFCs+HSC-2 groups). Black arrows indicated representative TRAP-positive multinucleated osteoclasts. (**e**) Total RAW264.7 cell number and (**f**) percentage of TRAP-positive RAW264.7 cells were quantified. Data were presented as mean ± SD of three independent experiments. **** *p* < 0.0001.

**Figure 3 cancers-17-01559-f003:**
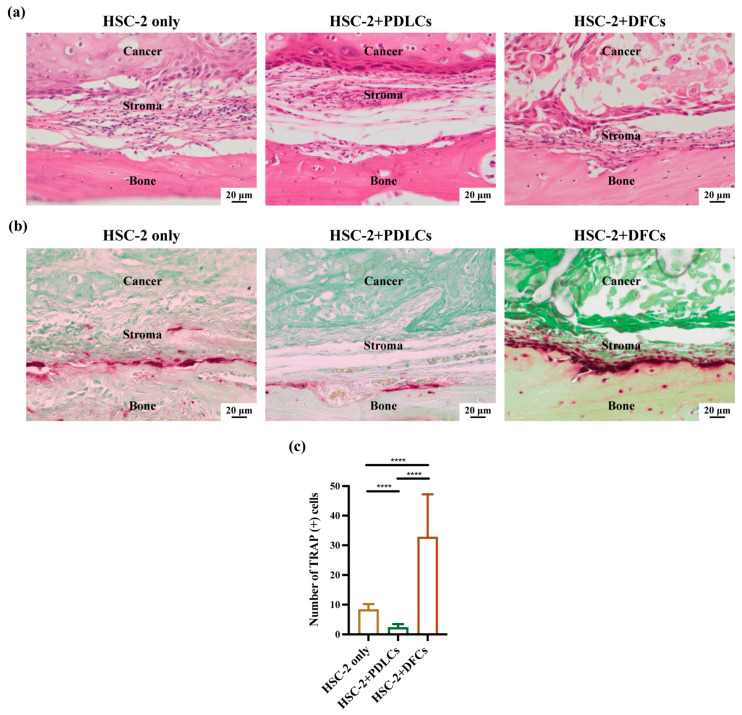
Effects of the crosstalk between DFCs/PDLCs and HSC-2 on bone surface invasion and osteoclast activation in well-differentiated OSCC in vivo. (**a**) Hematoxylin and eosin (HE) staining was performed to assess histological changes and the degree of bone resorption at the bone surface. (**b**) TRAP staining was conducted to identify TRAP-positive cells on the bone surface among the three groups (HSC-2, HSC-2+PDLCs, and HSC-2+DFCs). (**c**) Quantification of TRAP-positive cells on the bone surface across the three groups. Data were presented as mean ± SD from three independent experiments. **** *p* < 0.0001.

**Figure 4 cancers-17-01559-f004:**
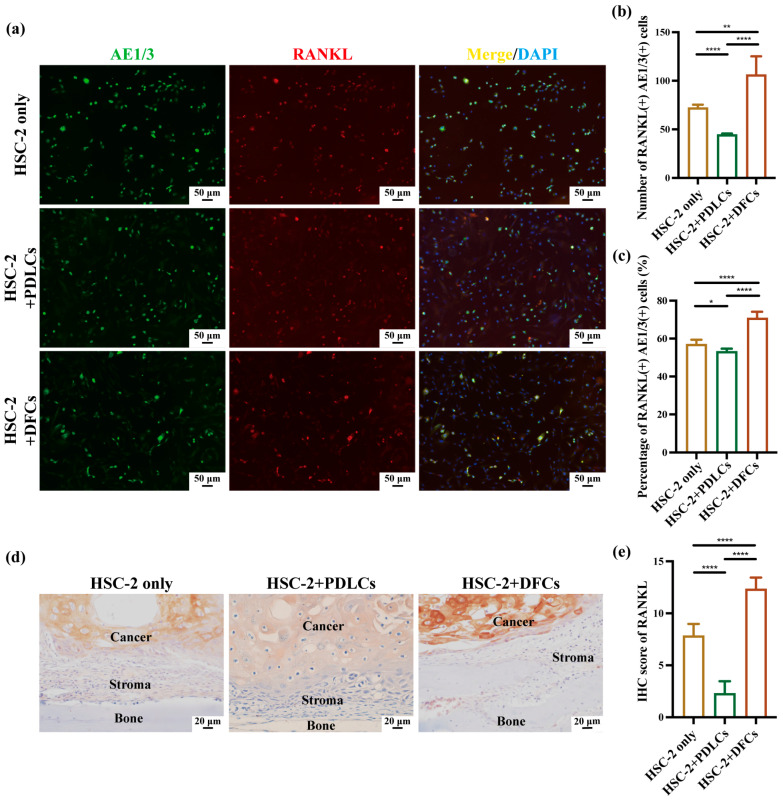
Effects of DFCs and PDLCs on RANKL expression on bone surface in OSCC in vitro and in vivo. (**a**) AE1/3 and RANKL immunofluorescence (IF) staining was conducted to assess RANKL expression in tumor cells in vitro. (**b**) Quantification of the number of RANKL(+)AE1/3(+) double-positive cells and (**c**) percentage of RANKL(+)AE1/3(+) double-positive cells across the three groups. Data were presented as mean ± SD from three independent experiments. * *p* < 0.05, ** *p* < 0.01, **** *p* < 0.0001. (**d**) Immunohistochemistry (IHC) staining was performed to visualize RANKL expression in cancer cells at the bone surface in OSCC in vivo. (**e**) RANKL expression was quantitatively analyzed using IHC scoring. Data were expressed as the mean ± SD from three independent experiments. **** *p* < 0.0001.

**Figure 5 cancers-17-01559-f005:**
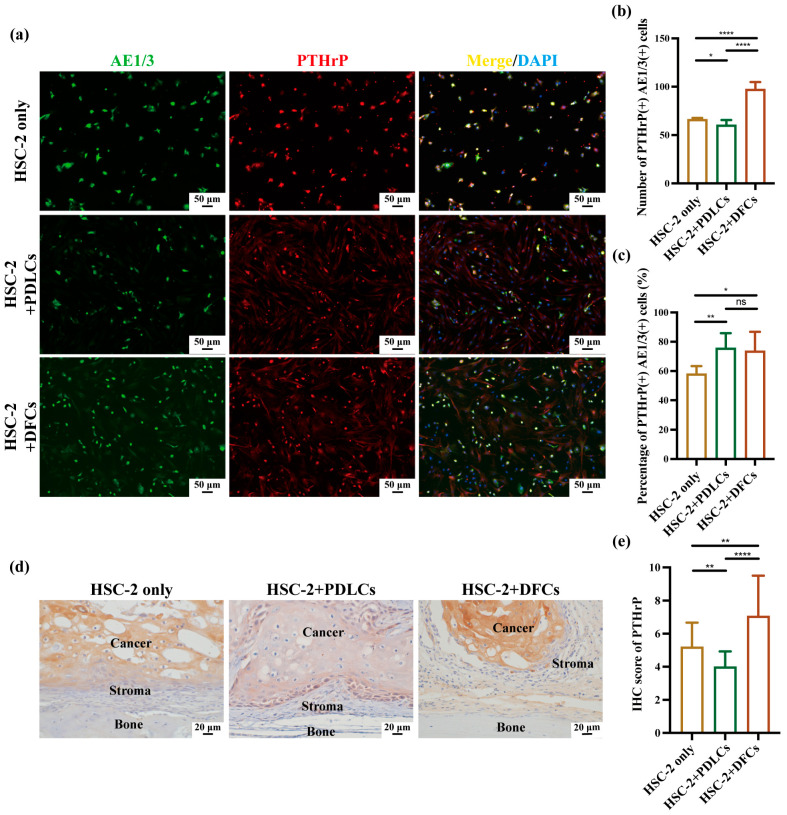
Effects of DFCs and PDLCs on PTHrP expression on bone surface in OSCC in vitro and in vivo. (**a**) AE1/3 and PTHrP double immunofluorescence (IF) staining was performed to assess PTHrP expression in tumor cells in vitro. (**b**) Quantification of the number of PTHrP(+)AE1/3(+) double-positive cells and (**c**) percentage of PTHrP(+)AE1/3(+) double-positive cells across the three groups. Data were presented as mean ± SD from three independent experiments. * *p* < 0.05, ** *p* < 0.01, **** *p* < 0.0001. (**d**) Immunohistochemistry (IHC) staining was performed to visualize PTHrP expression in cancer cells at the bone surface in OSCC in vivo. (**e**) Quantification of PTHrP expression using IHC score. Data were presented as mean ± SD from three independent experiments. ** *p* < 0.01, **** *p* < 0.0001.

**Figure 6 cancers-17-01559-f006:**
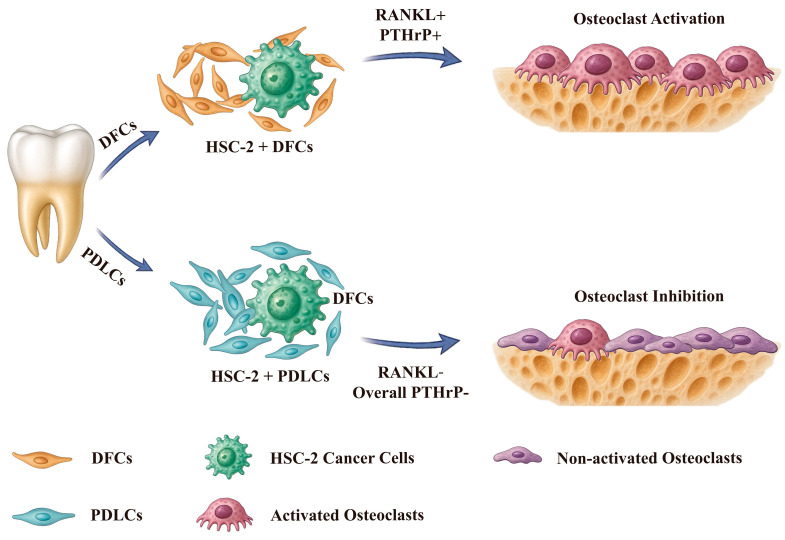
Schematic model illustrating the distinct regulatory roles of DFCs and PDLCs in osteoclast activation within the OSCC bone-invasive microenvironment. This diagram summarizes the cellular interactions observed in the current study. DFCs, derived from the dental follicle surrounding the unerupted crown and cervical region of the tooth, interact with HSC-2 cells to upregulate the expression of RANKL and PTHrP, thereby promoting osteoclast activation and bone resorption. In contrast, PDLCs, isolated from the periodontal ligament of the tooth apical region, suppress RANKL and overall PTHrP expression, resulting in decreased osteoclast activation and attenuated bone destruction. This model provides mechanistic insights into how different dental mesenchymal stromal cells distinctly modulate OSCC-associated bone resorption.

## Data Availability

The data presented in this study are available on reasonable request from the corresponding author. Due to ethical restrictions, the data are not publicly available.

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
