# Peer review of "Impacts of Dental Follicle Cells and Periodontal Ligament Cells on the Bone Invasion of Well-Differentiated Oral Squamous Cell Carcinoma"

_cancers, 2025, doi:10.3390/cancers17091559_

Round 1
Reviewer 1 Report
Comments and Suggestions for Authors
In this study, the authors propose that DFCs facilitate OSCC bone invasion by enhancing osteoclast activation through the RANKL and PTHrP signaling pathways, whereas PDLCs inhibit this osteolytic process by downregulating these same mediators. This may provide insights into the mechanisms by which stromal cells facilitate the tumor-invasive microenvironment. While the findings are intriguing, some aspects should be clarified more clearly.
1. What are the differences between DFCs and PDLCs? The expression of distinct cell markers should be examined to distinguish them more clearly. It is also recommended to provide a more detailed description of the procedures used to extract these two cell types. In addition, information regarding KUSA cells should be clearly described in the Materials and Methods section.
2. In the TRAP staining assay, how is the ratio of RAW264.7:DFCs/PDLCs:HSC-2 determined? If only two cell types are used—RAW264.7 and DFCs or PDLCs—what ratio is applied?
3. In Figure 1b, the staining intensity of PDLCs appears stronger than that of DFCs. It would be much better to provide quantitative analysis to support this observation. Moreover, in Figure 1c, both cell types exhibit osteogenic differentiation potential, although their ALP activity is lower than that of the KUSA cells. This suggests that both DFCs and PDLCs can differentiate into osteoblasts under appropriate environmental conditions. However, the subsequent data focus solely on osteoclast-related factors and do not further investigate osteogenic markers. Additionally, it is unclear why Alizarin Red staining is performed after two weeks of osteogenic induction, whereas ALP activity is measured after only five days of culture in osteogenic medium.
4. In Figure 2a, why does TRAP expression appear only in the RAW264.7 cell-only group? The cell density in the RAW264.7 + PDLCs group appears noticeably lower than in the other two groups. Are all four cell types—RAW264.7, DFCs, PDLCs, and HSC-2—cultured in the same medium? If so, does the co-culture condition affect cell viability? Moreover, how is the proliferating cell population identified—whether it is RAW264.7, PDLCs, or DFCs? The same question applies to the co-culture experiments involving HSC-2 cells. Are there any other methods to confirm that the proliferating population consists solely of RAW264.7 cells.The cell morphology in the PDLC co-culture groups appears quite different from that observed in the other groups.
5. "In lines 294–295, it is stated: 'To further confirm osteoclast activation, TRAP staining was applied to analyze RAW264.7 activation in vivo (Figure 3B).' However, the origin of RAW264.7 cells in this context is unclear. Were RAW264.7 cells co-injected with HSC-2 cells? Additionally, what is the rationale for injecting PDLCs and DFCs along with HSC-2 cells to establish the mouse model? PDLCs and DFCs are expected to reside in the tumor stroma naturally once the tumor aggregates form, so it is unclear why exogenous injection of these cells is necessary in this model.
6. In Figures 4a and 5a, the expression of RANKL and PTHrP is examined in HSC-2 cells with or without co-culture with PDLCs or DFCs. However, it is somewhat unclear whether RANKL is primarily expressed by HSC-2 cells or by the PDLCs/DFCs. Based on earlier results, co-culture of RAW264.7 cells with PDLCs or DFCs induces RAW264.7 differentiation or proliferation. With the addition of HSC-2 cells, it is unclear whether the expression of osteoclast-promoting factors is altered in PDLCs/DFCs, or whether PDLCs/DFCs modulate the expression profile of HSC-2 cells. This point requires further clarification. In addition, there appears to be a discrepancy in the HSC-2 + PDLCs group shown in Figures 4c and 5c. While the expression of RANK(+)AE1/3(+) cells is significantly decreased, the expression of PTHrP(+)AE1/3(+) cells is enhanced. This inconsistency requires further explanation.
7. In Figures 4d and 5d, RANKL and PTHrP staining appears to be confined to the tumor region, with little or no signal observed in the stromal region. Could additional experiments be performed to detect RANKL/PTHrP expression and osteoclast activation in serial sections or on the same slide to clarify their spatial relationship?
8. It would be helpful to include a brief schematic summarizing the interactions within the tumor–stroma–bone microenvironment.
Comments on the Quality of English LanguageIt would be better if it were revised by a native speaker.
Author Response
|
Comments 1: What are the differences between DFCs and PDLCs? The expression of distinct cell markers should be examined to distinguish them more clearly. It is also recommended to provide a more detailed description of the procedures used to extract these two cell types. In addition, information regarding KUSA cells should be clearly described in the Materials and Methods section. |
|
Response 1: Thank you for your valuable comments. Regarding the procedures for isolating DFCs and PDLCs, we have now provided a more detailed description in the Materials and Methods section. Although these protocols were previously well-established and cited in our earlier studies [8, 9, 26], we fully agree with your suggestion and have added a full explanation of the tissue processing and enzymatic digestion steps to ensure clarity and completeness (Page 3, Paragraph 4, Lines 125–149). [Updated text in the manuscript] As for the differences between DFCs and PDLCs, we would like to clarify that the aim of the immunofluorescence staining in our study was not to definitively distinguish these two cell types, but rather to confirm whether the isolated cells possess mesenchymal stem cell-like characteristics. Therefore, we used general mesenchymal stem cell markers (CD73, CD90, CD105) to evaluate their stemness, which is sufficient for our study's purpose. Finally, in response to your suggestion about the KUSA cell line, we have now added a description of KUSA cells in the Materials and Methods section (Page 4, Paragraph 2, Lines 154–157). [Updated text in the manuscript] “KUSA, a murine pre-osteoblastic cell line with fibrocyte-like morphology, was established from bone marrow stromal cells derived from primary femoral bone cultures of female C3H/He mice. This cell line was kindly provided by Dr. Akihiro Umezawa of Keio University (Tokyo, Japan).” We have also clarified the role of KUSA cells, which are mesenchymal stem cell-derived osteoblastic cells that exhibit mature osteoblast-like properties. They are capable of reproducible osteogenic differentiation both in vitro and in vivo [27]. Compared with other commonly used cell lines, KUSA cells possess a high and stable mineral deposition capacity, which makes them ideal as a positive control in osteogenic assays. Therefore, in this study, KUSA cells were used as a positive control for both the mineral deposition assay and the alkaline phosphatase (ALP) activity assay (Page 5, Paragraph 1, Lines 194–199) (Page 5, Paragraph 2, Lines 204–206). [Updated text in the manuscript] “KUSA cells exhibit mature osteoblast-like properties and are capable of consistent osteogenic differentiation both in vitro and in vivo [27]. Due to their robust and reproducible mineralization capacity, KUSA cells are widely used as a positive control in osteogenic assays. In this study, they were included as a positive control to benchmark the mineral deposition of DFCs and PDLCs.” “KUSA cells were used as a positive control in this assay due to their robust and reproducible osteogenic potential, particularly their high ALP activity during early stages of differentiation [27]. Their inclusion allowed us to benchmark the osteogenic response of DFCs and PDLCs under identical conditions.” |
|
Comments 2: In the TRAP staining assay, how is the ratio of RAW264.7:DFCs/PDLCs:HSC-2 determined? If only two cell types are used—RAW264.7 and DFCs or PDLCs—what ratio is applied? |
|
Response 2: Thank you for your insightful comment. The seeding ratios have been described in the Materials and Methods section (Page 5, Paragraph 3, Lines 210–212). In the triple co-culture of RAW264.7, DFCs/PDLCs, and HSC-2, a 3:3:1 ratio was used. This ratio was determined based on previous optimization studies [8,9,26] and adjusted according to the time each cell type takes to reach confluence in monoculture, to ensure synchronized growth and effective interaction in co-culture (Page 5, Paragraph 3, Lines 219–220). For dual co-culture (RAW264.7 with DFCs or PDLCs), a 3:3 ratio (approximately 1:1) was applied. [Updated text in the manuscript] “This seeding ratio was also adjusted according to the time each cell type takes to reach confluence in monoculture, to ensure synchronized growth in co-culture conditions.”
Comments 3: In Figure 1b, the staining intensity of PDLCs appears stronger than that of DFCs. It would be much better to provide quantitative analysis to support this observation. Moreover, in Figure 1c, both cell types exhibit osteogenic differentiation potential, although their ALP activity is lower than that of the KUSA cells. This suggests that both DFCs and PDLCs can differentiate into osteoblasts under appropriate environmental conditions. However, the subsequent data focus solely on osteoclast-related factors and do not further investigate osteogenic markers. Additionally, it is unclear why Alizarin Red staining is performed after two weeks of osteogenic induction, whereas ALP activity is measured after only five days of culture in osteogenic medium. Response 3: Thank you for your insightful comments. We agree that both DFCs and PDLCs exhibit osteogenic potential under appropriate conditions, as shown in Figure 1B and 1C. These experiments were primarily conducted to confirm that the stromal cells used in this study were functionally viable and capable of osteogenic differentiation. Interestingly, despite their osteogenic potential, these cells appeared to contribute to bone resorption in the tumor context, which we believe is one of the most important and thought-provoking aspects of this study. However, the core focus of this study was to investigate how these stromal cells contribute to osteoclast activation and bone invasion in the tumor microenvironment. Therefore, we did not further assess osteogenic markers such as Runx2, ALP, or OCN. We acknowledge this as a limitation and have added a statement to the Discussion section (Page 13, Paragraph 4, Lines 458–473). Regarding the experimental timeline, ALP activity was measured on Day 5 to assess early-stage differentiation, while Alizarin Red staining was performed after 3 weeks to evaluate late-stage matrix mineralization. This is in accordance with standard protocols [Gregory et al., Anal Biochem, 2004]. [Updated text in the manuscript] “Our data confirmed the osteogenic potential of both DFCs and PDLCs, as demonstrated by mineral deposition observed in Alizarin Red staining (Figure 1B) and ALP activity following osteogenic induction (Figure 1C). Although these assays qualitatively support osteogenic differentiation, we acknowledge that no quantitative image analysis (e.g., using ImageJ) was performed for Figure 1B, which may limit the objectivity of this result… …However, it is important to note that the primary aim of this study was to investigate the osteoclastogenic influence of stromal–tumor interactions, rather than to comprehensively assess the osteogenic profiles of DFCs and PDLCs. For this reason, we did not further evaluate molecular markers such as Runx2, ALP, or OCN. Future research should address this limitation by incorporating quantitative and molecular analyses to elucidate how these cells regulate the balance between bone formation and resorption in OSCC.”
Comments 4: In Figure 2a, why does TRAP expression appear only in the RAW264.7 cell-only group? The cell density in the RAW264.7 + PDLCs group appears noticeably lower than in the other two groups. Are all four cell types—RAW264.7, DFCs, PDLCs, and HSC-2—cultured in the same medium? If so, does the co-culture condition affect cell viability? Moreover, how is the proliferating cell population identified—whether it is RAW264.7, PDLCs, or DFCs? The same question applies to the co-culture experiments involving HSC-2 cells. Are there any other methods to confirm that the proliferating population consists solely of RAW264.7 cells. The cell morphology in the PDLC co-culture groups appears quite different from that observed in the other groups. Response 4: Thank you for your thoughtful and detailed comments. We would like to clarify the following points: 1) Culture Conditions: 2) TRAP Expression in RAW264.7 Alone Group: 3) Lower Cell Density in RAW264.7 + PDLCs Group: 4) Identification of Cell Types in Co-Culture: 5) Definition of Proliferating Population: 6) Biological Interpretation: We have now clarified these aspects in the revised manuscript (Page 8, Paragraph 5, Lines 318–328) (Page 9, Paragraph 2, Lines 338–352). [Updated text in the manuscript] “The impact of DFCs and PDLCs on RAW264.7 cells was assessed to evaluate their roles in osteoclast proliferation and differentiation... …Interestingly, the RAW264.7 + DFCs group exhibited the lowest percentage of TRAP-positive RAW264.7 cells, suggesting that DFCs predominantly promoted RAW264.7 proliferation rather than osteoclast differentiation (Figure 2C). In contrast, PDLCs facilitated the differentiation of RAW264.7 cells into osteoclasts.” “Notably, the RAW264.7 + PDLCs group showed a markedly reduced total cell count, despite being cultured under the same conditions and initial seeding density. This may reflect the inhibitory effect of PDLCs on RAW264.7 proliferation. Moreover, as terminal osteoclast differentiation is generally associated with reduced proliferative capacity, the increased differentiation observed in PDLC-containing groups may inherently result in fewer RAW264.7 cells overall [40]... …Compared to the other groups, the RAW264.7 + DFCs + HSC-2 group exhibited both the highest total RAW264.7 cell count and the highest percentage of TRAP-positive RAW264.7 cells, indicating that DFCs, in combination with HSC-2, enhanced both RAW264.7 proliferation and differentiation into osteoclasts (Figure 2E,F). Conversely, the RAW264.7 + PDLCs + HSC-2 group showed suppressed proliferation and differentiation of RAW264.7 cells into osteoclasts, further supporting the inhibitory role of PDLCs in osteoclastogenesis.”
|
|
Comments 5: "In lines 294–295, it is stated: 'To further confirm osteoclast activation, TRAP staining was applied to analyze RAW264.7 activation in vivo (Figure 3B).' However, the origin of RAW264.7 cells in this context is unclear. Were RAW264.7 cells co-injected with HSC-2 cells? Additionally, what is the rationale for injecting PDLCs and DFCs along with HSC-2 cells to establish the mouse model? PDLCs and DFCs are expected to reside in the tumor stroma naturally once the tumor aggregates form, so it is unclear why exogenous injection of these cells is necessary in this model. Response 5: Thank you for this important comment. We apologize for the lack of clarity. In the in vivo experiments, only HSC-2 cells and either DFCs or PDLCs were subcutaneously injected into the mice without RAW264.7 cells. RAW264.7 cells were not introduced, and the term “RAW264.7” in line 693 has been corrected to “osteoclasts” to avoid confusion (Page 10, Paragraph 1, Lines 357–358). Osteoclast precursors such as monocytes/macrophages can be recruited from the host in vivo and subsequently differentiate under the influence of tumor and stromal signals. As for the rationale behind co-injecting stromal cells with HSC-2, while stromal cells can indeed be recruited endogenously during tumor progression, we chose to directly introduce well-characterized DFCs or PDLCs in order to better control the composition of the tumor microenvironment. This ensured that the stromal component was present from the early stages of tumor formation, allowing us to directly compare the effects of these two types of stromal cells on osteoclastogenesis in vivo. Similar co-injection strategies have also been employed in previous studies to investigate the contribution of stromal cells to tumor-induced bone destruction [Suzuki et al., PLoS One, 2014; Nowlan et al., BMC Cancer, 2022]. [Updated text in the manuscript] “To further confirm osteoclast activation, TRAP staining was applied to analyze osteoclasts activation in vivo (Figure 3B). “
Comments 6: In Figures 4a and 5a, the expression of RANKL and PTHrP is examined in HSC-2 cells with or without co-culture with PDLCs or DFCs. However, it is somewhat unclear whether RANKL is primarily expressed by HSC-2 cells or by the PDLCs/DFCs. Based on earlier results, co-culture of RAW264.7 cells with PDLCs or DFCs induces RAW264.7 differentiation or proliferation. With the addition of HSC-2 cells, it is unclear whether the expression of osteoclast-promoting factors is altered in PDLCs/DFCs, or whether PDLCs/DFCs modulate the expression profile of HSC-2 cells. This point requires further clarification. In addition, there appears to be a discrepancy in the HSC-2 + PDLCs group shown in Figures 4c and 5c. While the expression of RANK(+)AE1/3(+) cells is significantly decreased, the expression of PTHrP(+)AE1/3(+) cells is enhanced. This inconsistency requires further explanation. Response 6: Thank you for your insightful comments. We address your questions as follows: 1. Regarding the source of RANKL and PTHrP expression in Figures 4 and 5: [Updated text in the manuscript] “Furthermore, immunohistochemical analysis confirmed that RANKL and PTHrP expression was predominantly localized within AE1/3-positive OSCC regions, with minimal staining observed in the surrounding stroma… …Our findings build upon this understanding by highlighting how tumor–stromal interactions, particularly involving DFCs and PDLCs, dynamically modulate the expression of these key mediators in the tumor microenvironment.” 2. Regarding the apparent discrepancy in the PDLC co-culture group in Figure 5: This reflects a phenomenon of selective enrichment of functional subpopulations, as reported in previous studies of stromal-induced tumor heterogeneity via TGF-β or BMP signaling [Wendt et al., Oncogene, 2009; Hill et al., Cancer Res, 2005]. We have clarified these interpretations in the revised manuscript (Page 14, Paragraph 3, Lines 474–486). [Updated text in the manuscript] “Quantitative immunohistochemistry revealed that co-culture with DFCs significantly increased the number, percentage, and IHC scores of RANKL(+)AE1/3(+) and PTHrP(+)AE1/3(+) cells, suggesting a broad upregulation of osteoclast-activating signals by DFCs… …This paradox may reflect a global inhibitory effect of PDLCs on HSC-2 proliferation and protein synthesis, leading to a reduced tumor mass while selectively enriching PTHrP-expressing subpopulations. Such stromal cell-induced phenotypic selection has been previously reported in tumor heterogeneity mediated by TGF-β and BMP signaling [41,42].“
Comments 7: In Figures 4d and 5d, RANKL and PTHrP staining appears to be confined to the tumor region, with little or no signal observed in the stromal region. Could additional experiments be performed to detect RANKL/PTHrP expression and osteoclast activation in serial sections or on the same slide to clarify their spatial relationship? Response 7: Thank you for your thoughtful and valuable suggestion. As noted, our immunohistochemical staining showed that both RANKL and PTHrP were predominantly localized within AE1/3-positive OSCC tumor regions, with minimal to no signal detected in the surrounding stromal areas (Figures 4D and 5D). This finding indicates that HSC-2 cells likely serve as the principal source of osteoclast-activating factors in our model. This observation is supported by previous studies demonstrating that OSCC and other cancer cells can directly produce RANKL and PTHrP, which activate osteoclasts through autocrine and paracrine mechanisms [Mori et al., Oral Oncol.2014; Guise et al., Clin Cancer Res. 2006]. Among these, tumor cell-derived secretion has been shown to play a dominant role in driving osteoclastogenesis. We agree that investigating the spatial relationship between these osteolytic factors and osteoclast activation would add further depth to our understanding. In the present study, we used single-marker staining due to technical limitations, but acknowledge that multiplex or serial-section staining could provide clearer spatial correlations. To address this limitation, we have now added a statement in the Discussion section indicating that future studies will include dual immunostaining or consecutive serial sections to more precisely define the spatial interplay between tumor-derived RANKL/PTHrP expression and osteoclast localization(Page 15, Paragraph 3, Line 522-537). [Updated text in the manuscript] “Despite these insights, several limitations should be acknowledged… …Fourth, while our immunohistochemical results suggested that RANKL and PTHrP expression was largely confined to AE1/3-positive OSCC tumor regions, their precise spatial relationship with osteoclast activation remains unclear. Future studies employing dual-marker staining or serial-section analysis will be needed to map these interactions more accurately… …These limitations should be addressed in future investigations to further validate and expand upon our findings.”
Comments 8: It would be helpful to include a brief schematic summarizing the interactions within the tumor–stroma–bone microenvironment. Response 8: Thank you very much for your constructive suggestion. In response, we have added a schematic illustration (now included as Figure 6) to visually summarize the proposed interactions within the OSCC tumor–stroma–bone microenvironment. This diagram emphasizes the contrasting regulatory effects of DFCs and PDLCs on osteoclast activity through their interaction with HSC-2 cells. Specifically, DFCs enhance RANKL and PTHrP expression in HSC-2 cells, thereby promoting osteoclastogenesis and bone resorption. In contrast, PDLCs suppress RANKL and reduce overall PTHrP expression, contributing to the inhibition of osteoclast-mediated bone destruction. We believe this schematic helps convey the mechanistic insights of our study more intuitively (Page 15, Paragraph 1, Line 503-512). [Updated text in manuscript]
Response to Comments on the Quality of English Language |
|
Point : It would be better if it were revised by a native speaker. |
|
Response : Thank you for your suggestion. We have carefully revised the entire manuscript to improve clarity, grammar, and overall readability. To ensure linguistic accuracy, the manuscript has been thoroughly edited by a native English speaker with scientific writing experience. We hope that the revised version meets the journal’s language standards.
|
|
Additional clarifications |
|
We would also like to express our sincere gratitude to Reviewer 1 for the thorough and thoughtful evaluation of our manuscript. We truly appreciate the many insightful comments and constructive suggestions provided. While we acknowledge that certain aspects of the study may not fully meet the reviewer’s expectations due to the current limitations, we are committed to improving the rigor and comprehensiveness of our future work. Regarding the co-culture ratio used in this study (RAW264.7:DFCs/PDLCs:HSC-2 = 3:3:1), this was determined based on preliminary optimization experiments and prior studies from our group (Refs. 8, 9, 26). The ratio was selected to ensure synchronized confluence and adequate cellular interactions while minimizing contact inhibition, and it was applied consistently across all relevant experiments. As for the reviewer’s suggestion to include quantitative analysis of mineral deposition (e.g., in Figure 1B), we agree that this would strengthen the objectivity of the data. However, given that the core aim of our study is to explore osteoclast-mediated mechanisms, we did not pursue this analysis to avoid deviating from the main focus. We have now clarified this rationale in the revised Discussion section and acknowledged the lack of mineral quantification as a limitation. We sincerely appreciate this feedback and will integrate such quantitative assessments in future studies.
|

Reviewer 2 Report
Comments and Suggestions for Authors
2.5 Mineralization Assay by Alizarin Red
Please indicate the cell number or confluency at the time of assay.
2.8 Construction of Animal Model
Please specify the age (in weeks) of the mice used in the experiment.
Figure 1(a): AE1/3 & Vimentin, CD73 & CD105, and CD90 & CD73
Please provide quantification of the fluorescence signals.
Author Response
|
Comments 1: 2.5 Mineralization Assay by Alizarin Red |
|
Response 1: Thank you for your helpful comment. In response, we have now added the specific seeding density and confluency criteria to the revised Materials and Methods section (Page 4, Paragraph 5, Lines 186–192). DFCs, PDLCs, and KUSA were seeded at 1 × 10⁵ cells per well in 12-well plates with 2 mL of α-MEM supplemented with 10% FBS. When cells reached approximately 90% confluency, the medium was replaced with osteogenic induction medium. [updated text in the manuscript] “To evaluate the osteogenic differentiation potential, DFCs, PDLCs and KUSA were seeded at a density of 1 × 10⁵ cells per well in 12-well plates and cultured in α-MEM supplemented with 10% fetal bovine serum (FBS, Biowest, Nuaillé, France), totally 2 mL. When the cells reached approximately 90% confluency, the culture medium was replaced with 2 mL of osteogenic induction medium consisting of α-MEM, 10% FBS, 10 mM β-glycerophosphate (Sigma-Aldrich, St. Louis, MO, USA), and 0.05 mM L-ascorbic acid (Sigma-Aldrich, St. Louis, MO, USA).“
|
|
Comments 2: 2.8 Construction of Animal Model |
|
Response 2: Thank you for your helpful comment. We confirm that the age of the mice—7-week-old female BALB/c nu-nu mice—was already described in the original manuscript. To ensure clarity, we have retained and highlighted this information in the revised Materials and Methods section (Page 5, Paragraph 5, Line 235-238). [updated text in the manuscript] “Following anesthesia, a total of 200 μL cell suspension (HSC-2, 1 × 10⁶, 100 μL; DFCs/PDLCs, 3 × 10⁶, 100 μL) was carefully injected into the subcutaneous tissue over the skull of 15 healthy 7-week-old female BALB/c nu-nu mice (15 g; Shimizu Laboratory Supplies Co., Ltd).”
Comments 3: Figure 1(a): AE1/3 & Vimentin, CD73 & CD105, and CD90 & CD73 Response 3: Thank you for your valuable comment. Figure 1A was designed to qualitatively characterize the mesenchymal identity and cellular composition of DFCs and PDLCs, rather than to compare fluorescence intensity between groups. As such, we did not perform signal quantification. Both cell types displayed spindle-shaped morphology, were positive for Vimentin and CD73, and negative for AE1/3, CD90, and CD105—confirming their mesenchymal characteristics and lack of epithelial contamination. Although CD90 and CD105 are commonly used mesenchymal stem cell markers, their expression can vary in dental-derived stromal cells depending on donor variability and culture conditions, which aligns with our findings [Yagyuu et al., 2015; Seo et al., 2004]. This clarification has been added to the legend of Figure 1A (Page 7, Paragraph 3, Line 298-300). [updated text in the manuscript] “As the primary aim of these immunofluorescence assays was to confirm cell identity and mesenchymal features, fluorescence intensity was not quantitatively analyzed.”
|
|
Response to Comments on the Quality of English Language |
|
Point : The English is fine and does not require any improvement. |
|
Response : Thank you for your feedback. We are pleased to know that the English in our manuscript meets the journal’s standards. We appreciate your positive evaluation.
|
|
Additional clarifications |
|
We would like to thank Reviewer 2 again for the valuable suggestions. In addition to providing point-by-point responses, we have carefully reviewed the entire manuscript and made several revisions to enhance clarity and transparency. These include the addition of key methodological details (e.g., cell seeding density, mouse age) and clarification of the rationale behind selected assays—such as the non-quantitative nature of Figure 1A immunofluorescence, which aimed to confirm cell identity rather than assess marker expression levels. We sincerely appreciate your time and thoughtful evaluation, and we hope these refinements have improved the scientific rigor and overall presentation of our work. |

Reviewer 3 Report
Comments and Suggestions for Authors
Suggestion for authors:
Introduction:
Concern: No previous studies have been cited dealing with the relationship between DFCs and PDLCs and OSCC carcinoma.
Suggestion for Manuscript Improvement: If any, disclose previous studies that have investigated the impact of DFCs and PDLCs on bone invasion in OSCC carcinoma such as the presence of these cell lines in the microenviroment of tumor front, their prognostic impact. Otherwise declare that the topic is unexlopred.
Materials and Methods:
1. Sample selection:
Concern: The authors state that “Primary DFCs and PDLCs were isolated from patients, with cells collected from im-113 pacted teeth. These cells were obtained from two different patients and confirmed to be 114 lesion-free by radiological and oral surgical examinations.” However no additional clinical information is provided
Suggestion for Manuscript Improvement: Please provide more detailed clinical information on healthy donors . Clinical parameters such as age, presence/ absence of relevant clinical conditions, drugs that could alter results should be disclosed. In addition, disclose if the two healthy donors had superimposable baseline characteristics and how they were selected.
2. Statistical analysis:
Concern: The authors state that “Cell-based experiments were conducted in triplicate, while animal experiments were repeated four times per group. Data were expressed as mean ± SD and analyzed using the Kruskal-Wallis test followed by the Steel-Dwass test. A P-value < 0.05 was considered statistically significant.” No justification is provided for the number of repetition of experiments or for the choice of statistical test assuming that data refer to non parametric population.
Suggestion for Manuscript Improvement: For a better readability authors should disclose why they adopted this protocol of repetition ( i.e to achieve sufficient numerosity and statistical power) and why they chose a non parametric test ( i.e after checking distribution of data)
Discussion and conclusions:
1. DFCs and OSCC bone invasion:
Concern: DFCs are more abundant and active during tooth eruption. Oral squamous cell carcinoma has its peak of incidence in the 6th 7th decade of life. Do authors have data on the role and activity of DFCs in adults that could impact the progression of OSCC ?
Suggestion for Manuscript Improvement: Please describe more in detail the impact of DFCs on OSCC progression in adults and describe the design of possible future investigations that may corroborate your results
2. PDLCs and OSCC bone invasion:
Concern: Do authors have data on the role and activity of PDLCs that could impact the progression of OSCC ?
Suggestion for Manuscript Improvement: If any, please include previous studies that investigated the impact of periodontal ligament cells on OSCC bone invasion and progression.
3. Limitations:
Concern: The authors do not mention potential limitation of their study
Suggestion for Manuscript Improvement: Please describe the limitations of the present study ( i.e results from in vitro and in vivo experiments that should be confirmed in real clinical settings) , and suggest potential future lines of investigation for future studies.
Author Response
|
Comments 1: No previous studies have been cited dealing with the relationship between DFCs and PDLCs and OSCC carcinoma. Please cite any relevant studies, or state clearly that the topic remains unexplored. |
|
Response 1: Thank you very much for your insightful comment. In response, we have clarified the current state of research regarding the relationship between DFCs, PDLCs, and OSCC in the revised Introduction. Specifically, we have cited our previous study [9], which demonstrated that PDLCs inhibited osteoclast activation and tumor-induced bone destruction in an OSCC xenograft model, suggesting their potential protective role in OSCC-associated bone resorption (Page 3, Paragraph 2, Lines 104–108). [updated text in the manuscript] “In our previous study using an OSCC xenograft model, PDLCs were found to inhibit osteoclast activation and reduce tumor-induced bone destruction, suggesting a potential suppressive role of PDLCs in the context of cancer-associated bone resorption [9]. However, the underlying mechanisms by which PDLCs modulate OSCC-induced bone invasion remain largely unexplored and warrant further investigation.” In contrast, while DFCs are well recognized for their role in physiological bone remodeling and tooth eruption, their involvement in cancer-related bone resorption, particularly in OSCC, has not been previously elucidated. We have explicitly stated this knowledge gap in the Introduction (Page 2, Paragraph 4, Lines 93-96). [updated text in the manuscript] “Despite their physiological relevance, the potential involvement of DFCs in OSCC-associated bone destruction has not been well characterized in the context of cancer-induced bone resorption, especially in OSCC, leaving an important gap in current knowledge.” To our knowledge, this is one of the first studies to directly compare the differential effects of DFCs and PDLCs on OSCC-induced bone invasion (Page 3, Paragraph 3, Lines 120–122). [updated text in the manuscript] “To our knowledge, this is among the first studies to systematically compare the distinct effects of DFCs and PDLCs on OSCC-induced bone resorption.”
|
|
Comments 2: [Materials and Methods----Sample selection] Concern: The authors state that “Primary DFCs and PDLCs were isolated from patients, with cells collected from impacted teeth. These cells were obtained from two different patients and confirmed to be lesion-free by radiological and oral surgical examinations.” However no additional clinical information is provided. Suggestion for Manuscript Improvement: Please provide more detailed clinical information on healthy donors. Clinical parameters such as age, presence/absence of relevant clinical conditions, drugs that could alter results should be disclosed. In addition, disclose if the two healthy donors had superimposable baseline characteristics and how they were selected. |
|
Response 2: Thank you for your valuable comment. In response, we have expanded the description of sample selection and donor characteristics in the revised Materials and Methods section (Page 3, Paragraph 4, Lines 125–134). Primary DFCs and PDLCs were isolated from two healthy, non-smoking adult male patients (born in 1999 and 2000) undergoing third molar extraction at the Department of Oral and Maxillofacial Surgery, Okayama University. Both donors were free of systemic diseases, not on any long-term medications, and showed no clinical or radiographic signs of oral pathology. These characteristics were confirmed through preoperative examinations. Identifying personal data are omitted for privacy reasons but are retained in the original informed consent documentation. To isolate DFCs, we used completely impacted third molars, from which dental follicle tissues were harvested. These tissues are located around the crown and cervical region and are only present before eruption. In contrast, PDLCs were collected from the periodontal ligament along the root surface of partially or fully erupted third molars. All extractions were performed under standard clinical indications, and both donors exhibited comparable baseline profiles. Tissue acquisition followed a convenience sampling strategy based on eligibility and specimen availability. We hope these additions adequately address the reviewer’s concerns. [updated text in the manuscript] “Primary DFCs and PDLCs were isolated from two healthy, non-smoking adult male patients who underwent third molar extraction at the Department of Oral and Maxillofacial Surgery, Okayama University. Both donors were confirmed to be free of systemic disease, long-term medication use, and oral pathological conditions, as determined by radiographic and clinical examination. Following informed consent and based on the protocol of our previous studies, the samples were collected under sterile conditions. Specifically, DFCs were harvested from the dental follicle tissues surrounding the crown and cervical regions of completely impacted, unerupted third molars, as these structures are only present prior to eruption. In contrast, PDLCs were isolated from the periodontal ligament tissue along the root surface of partially or fully erupted third molars [8,9,26].”
Comments 3: [Materials and Methods----Statistical analysis] Concern: The authors state that “Cell-based experiments were conducted in triplicate, while animal experiments were repeated four times per group. Data were expressed as mean ± SD and analyzed using the Kruskal-Wallis test followed by the Steel-Dwass test. A P-value < 0.05 was considered statistically significant.” No justification is provided for the number of repetition of experiments or for the choice of statistical test assuming that data refer to non parametric population. Suggestion for Manuscript Improvement: For a better readability authors should disclose why they adopted this protocol of repetition ( i.e to achieve sufficient numerosity and statistical power) and why they chose a non parametric test ( i.e after checking distribution of data) Response 3: Thank you for this helpful suggestion and for pointing out the inconsistency in our previous description. We sincerely apologize for the earlier error regarding the number of repetitions in the animal experiments. The correct information is as follows: each group contained five mice (n = 5 per group), and the animal experiment was independently repeated three times, not four. This has been corrected in the revised Materials and Methods section. The sample size of five mice per group was chosen based on previous literature and to ensure sufficient statistical power while respecting ethical considerations. This approach is consistent with the 3Rs principles (Replacement, Reduction, and Refinement), and provides adequate power to detect biologically meaningful differences while minimizing animal use (Page 5, Paragraph 5, Lines 235–243). [updated text in the manuscript] “Following anesthesia… …15 healthy 7-week-old female BALB/c nu-nu mice (15 g; Shimizu Laboratory Supplies Co., Ltd). A sample size of five animals per group (n = 5) was chosen based on standard practices in previous studies and ethical considerations. This number balances statistical power and biological reproducibility with the need to minimize animal use in accordance with the 3Rs principle (Replacement, Reduction, and Refinement). It was sufficient to detect significant differences across experimental conditions while adhering to institutional animal care guidelines.” In cell-based assays, experiments were conducted in biological triplicates to ensure reproducibility and control inter-experimental variability (Page 6, Paragraph 5, Lines 277–278). [updated text in the manuscript] “Cell-based experiments were conducted in biological triplicate, while animal experiments were repeated three times per group.” To determine the appropriate statistical method, we first assessed data normality using the Shapiro–Wilk test. Since some datasets did not meet the criteria for normal distribution, we adopted non-parametric methods: the Kruskal–Wallis test for group comparisons, followed by the Steel–Dwass post hoc test for multiple pairwise comparisons. These tests are suitable for small sample sizes and non-normally distributed data. All results are expressed as mean ± SD, and P < 0.05 was considered statistically significant. These justifications have now been incorporated into the revised Statistical Analysis section (Page 6, Paragraph 5, Lines 281–286). [updated text in the manuscript] “To determine the appropriate statistical method, the Shapiro–Wilk test was first performed to assess the normality of data distributions. As some datasets did not follow a normal distribution, non-parametric tests were chosen. Specifically, the Kruskal–Wallis test was used for overall comparisons, followed by the Steel–Dwass post hoc test for multiple pairwise analyses. All data were presented as mean ± SD, and a P-value < 0.05 was considered statistically significant.”
Comments 4: [Discussion and conclusions----DFCs and OSCC bone invasion] Response 4: Thank you for raising this insightful comment. We agree that DFCs are typically associated with the tooth eruption process and are more abundant during developmental stages. However, several recent studies have demonstrated that functional DFC-like stromal populations can still be isolated from the follicular tissues of impacted third molars in adults. In our study, DFCs were derived from adult patients (ages 22 and 24) undergoing third molar extraction. These cells exhibited mesenchymal-like morphology and osteoclast-activating potential, consistent with previous reports that adult DFCs retain osteogenic and immunomodulatory functions under specific environmental stimuli, such as inflammation or tumor-derived cytokines (Honda et al., 2010; Edamatsu et al., 2005; Zhang et al., 2019) [44–46]. To clarify this point, we have revised the Discussion to emphasize the relevance of adult DFCs in OSCC bone invasion (Page 14, Paragraph 5, Lines 496–502). [updated text in the manuscript] “Moreover, although DFCs are best known for their role during tooth eruption and are more prevalent in younger individuals, recent studies confirm the presence of functional DFCs even in adult impacted molars [45,46]. In our study, DFCs were derived from adults (ages 22 and 24) and retained osteoclast-activating potential and MSC-like features. This supports the idea that adult DFCs can respond to inflammatory or tumor-derived cues and actively modulate bone homeostasis [45,47]. Nonetheless, future studies should explore age- and stage-dependent variation in DFC function.” We fully acknowledge that the biological activity of DFCs may vary with donor age and clinical background. Future studies should investigate how age-related or inflammatory microenvironments influence DFC behavior and contribution to bone invasion. We have added this direction to the Discussion as well (Page 16, Paragraph 2, Lines 538–546). [updated text in the manuscript] “Future studies should aim to elucidate the precise molecular circuits through which DFCs and PDLCs regulate osteoclastogenesis in the OSCC microenvironment. To achieve this, a combination of cytokine profiling, single-cell transcriptomic analysis, and functional validation in age-matched in vivo models will be essential. Moreover, investigating how patient-specific factors, such as donor age or inflammatory status, influence the behavior of these stromal cells may provide further insight into their context-dependent roles. These approaches will help lay the groundwork for developing stroma-targeted therapeutic strategies to attenuate tumor-induced bone destruction and improve clinical outcomes in OSCC patients.”
Comments 5: [Discussion and conclusions----PDLCs and OSCC bone invasion] Response 5: Thank you for your thoughtful comment. In our previous study (Ref. [9]), we utilized a xenograft mouse model of oral squamous cell carcinoma (OSCC) to investigate the effects of various oral stromal cell populations on tumor-induced bone resorption. That study revealed that gingiva-derived stromal cells (G-SCs) promoted osteoclastogenesis and bone destruction, whereas periodontal ligament-derived stromal cells (P-SCs)—which correspond to the PDLCs used in the present study—exerted an inhibitory effect on both osteoclast activation and tumor-associated bone loss. These earlier findings are consistent with our current results, which demonstrate that PDLCs suppress the expression of RANKL and PTHrP in HSC-2 cells and inhibit osteoclastogenesis in both in vitro and in vivo settings. Taken together, these data support the hypothesis that PDLCs may serve as protective stromal components within the OSCC microenvironment by attenuating osteoclast-mediated bone invasion. We have now cited this prior work in the revised Introduction (Page 2, Paragraph 3, Lines 75–78)(Page 3, Paragraph 2, Lines 104–107) and Discussion (Page 13, Paragraph 3, Lines 451–457) to highlight the relevance and continuity of our research. We appreciate the reviewer’s suggestion, which helped clarify the scientific context and rationale for our investigation. [updated text in the manuscript] “Previous studies, including our own, have demonstrated that stromal cell components play a critical role in OSCC bone invasion by modulating tumor invasiveness, epithelial-mesenchymal transition (EMT), and the expression of RANKL and parathyroid hormone-related peptide (PTHrP) [8,9].” “In our previous study using an OSCC xenograft model, PDLCs were found to inhibit osteoclast activation and reduce tumor-induced bone destruction, suggesting a potential suppressive role of PDLCs in the context of cancer-associated bone resorption [9].” “Interestingly, in our previous study using a similar xenograft model, we found that gingival stromal cells (G-SCs) promoted osteolysis, whereas periodontal ligament stromal cells (P-SCs)—which correspond to the PDLCs used in the current study—exhibited an inhibitory effect on OSCC-induced bone resorption [9]. This finding is consistent with the present results and suggests that PDLCs may play a protective role in the tumor–bone microenvironment by attenuating osteoclast activation.”
Comments 6: [Discussion and conclusions----Limitations] Response 6: Despite the novel findings presented, several limitations of this study should be acknowledged. First, although DFCs and PDLCs demonstrated mineralization capacity and early osteogenic markers, quantitative analysis of mineral deposition (e.g., using ImageJ) was not performed. This may limit the objectivity of osteogenic evaluation. Second, molecular profiling of osteogenic markers such as Runx2, ALP, and OCN between DFCs and PDLCs was not further explored, as the primary aim of the study was to investigate osteoclast-related mechanisms. Third, while DFCs were derived from adult patients (ages 22 and 24), the study did not stratify cell behavior across different age groups or tumor stages. Fourth, the in vitro and subcutaneous xenograft models used here, though widely adopted in OSCC research, may not fully replicate the complex bone-tumor microenvironment observed in clinical settings. Finally, spatial resolution between osteoclast-activating factors (RANKL, PTHrP) and osteoclast localization was not directly demonstrated using multiplex staining or serial sections. We have now cited this prior work in the revised Discussion (Page 15, Paragraph 3, Lines 522–537) [updated text in the manuscript] “Despite these insights, several limitations should be acknowledged. First, mineral deposition was not quantitatively analyzed (e.g., using ImageJ), which may limit the objectivity of the osteogenic potential assessment… …These limitations should be addressed in future investigations to further validate and expand upon our findings.” To address these limitations, future studies should incorporate high-resolution techniques such as dual immunostaining or spatial transcriptomics to precisely map the spatial relationships within the tumor–stroma–bone microenvironment. Additionally, systematic evaluation of age-dependent stromal cell responses, combined with cytokine profiling and single-cell transcriptomics, may elucidate how patient-specific factors (e.g., age, inflammation) modulate the osteoclastogenic potential of DFCs and PDLCs. These investigations will not only deepen mechanistic understanding but also provide translational insights into stroma-targeted interventions for OSCC bone invasion. We have now cited this prior work in the revised Discussion (Page 16, Paragraph 2, Lines 538–546) [updated text in the manuscript] “Future studies should aim to elucidate the precise molecular circuits through which DFCs and PDLCs regulate osteoclastogenesis in the OSCC microenvironment. To achieve this, a combination of cytokine profiling, single-cell transcriptomic analysis, and functional validation in age-matched in vivo models will be essential. Moreover, investigating how patient-specific factors, such as donor age or inflammatory status, influence the behavior of these stromal cells may provide further insight into their context-dependent roles. These approaches will help lay the groundwork for developing stroma-targeted therapeutic strategies to attenuate tumor-induced bone destruction and improve clinical outcomes in OSCC patients.”
|
|
Response to Comments on the Quality of English Language |
|
Point : The English is fine and does not require any improvement. |
|
Response : Thank you very much for your positive feedback on the language quality. We truly appreciate your recognition. To further enhance clarity and fluency, we have nonetheless carefully re-reviewed and polished the manuscript during this revision.
|
|
Additional clarifications |
|
We would like to once again express our sincere appreciation to Reviewer 3 for the thoughtful comments and valuable suggestions, which have greatly contributed to improving the quality, clarity, and scientific rigor of our manuscript. In addition to the point-by-point responses and corresponding textual revisions, we would like to offer the following clarifications:
We hope these clarifications, along with the comprehensive revisions made throughout the manuscript, adequately address the reviewer’s concerns and strengthen the overall impact of our study. |

Reviewer 4 Report
Comments and Suggestions for Authors
The study aims to investigate the effects of dental follicle cells (DFCs) and periodontal ligament cells (PDLCs) on oral squamous cell carcinoma (OSCC) bone invasion and the underlying mechanisms. The study is interesting and well-structured. The manuscript is well-written. Some observations can be considered in further revisions.
- The type of the study should be declared in the title, abstract, and methods section.
- The abstract can be divided into sections: "aim, methods, results, conclusion"
- The sample size for both the in vivo and in vitro sections can be more clearly defined.
- A paragraph should be added at the end of the discussion section, where the authors describe limitations and considerations observed from their experience, to be helpful for future studies for the progression of this topic and better interpretation of the results.
Author Response
|
Comments 1: The study aims to investigate the effects of dental follicle cells (DFCs) and periodontal ligament cells (PDLCs) on oral squamous cell carcinoma (OSCC) bone invasion and the underlying mechanisms. The study is interesting and well-structured. The manuscript is well-written. Some observations can be considered in further revisions. - The type of the study should be declared in the title, abstract, and methods section. |
|
Response 1: Thank you very much for your positive evaluation and constructive comments. We greatly appreciate your kind encouragement. In response to your suggestion regarding the declaration of the study type, we agree that clearly identifying the nature of the study enhances transparency and improves reader understanding. While we chose to retain the original title for clarity and brevity, we have revised the Abstract to explicitly state that this is an experimental study incorporating both in vitro and in vivo approaches (Page 1, Paragraph 2, Lines 41–44). Furthermore, the structure of the Materials and Methods section already delineates the specific in vitro and in vivo experiments performed, which we believe clearly communicates the study design. We hope this clarification appropriately addresses your concern. [updated text in the manuscript] “This experimental study combined in vitro co-culture systems and an in vivo xenograft mouse model to investigate the distinct effects of DFCs and PDLCs on OSCC-induced bone resorption and elucidate the underlying mechanisms.”
|
|
Comments 2: The abstract can be divided into sections: "aim, methods, results, conclusion" |
|
Response 2: Thank you for your valuable suggestion. We fully agree that a structured abstract format with clearly delineated sections such as Aim, Methods, Results, and Conclusion can improve clarity and facilitate quick understanding. However, as per the official formatting guidelines of Cancers, the journal recommends a single-paragraph unstructured abstract. Therefore, to comply with these editorial standards, we have respectfully retained the original unstructured format. That said, we have carefully ensured that all essential components—namely, the study’s aim, methods, key findings, and conclusion—are presented in a logical and coherent flow. We hope this format remains acceptable and appreciate your kind understanding.
Comments 3: The sample size for both the in vivo and in vitro sections can be more clearly defined. Response 3: Thank you for your helpful comment. In the revised manuscript, we have clarified the sample sizes used in both the in vivo and in vitro experiments to improve transparency and reproducibility. For the in vivo study, a total of 15 healthy 7-week-old female BALB/c nu-nu mice (15 g) were used. These animals were randomly assigned into three groups (n = 5 per group): HSC-2 only, HSC-2 + DFCs, and HSC-2 + PDLCs. A sample size of five animals per group (n = 5) was chosen based on standard practices in previous studies and ethical considerations. This number balances statistical power and biological reproducibility with the need to minimize animal use in accordance with the 3Rs principle (Replacement, Reduction, and Refinement). It was sufficient to detect significant differences across experimental conditions while adhering to institutional animal care guidelines. Information has been clearly described in the Materials and Methods section (Page 5, Paragraph 5, Lines 235–247). [updated text in the manuscript] “Following anesthesia, a total of 200 μL cell suspension (HSC-2, 1 × 10⁶, 100 μL; DFCs/PDLCs, 3 × 10⁶, 100 μL) was carefully injected into the subcutaneous tissue over the skull of 15 healthy 7-week-old female BALB/c nu-nu mice (15 g; Shimizu Laboratory Supplies Co., Ltd)… …Mice had free access to food and water and were randomly assigned to three groups (n = 5 per group): HSC-2 only, HSC-2 + DFCs, and HSC-2 + PDLCs.” For the in vitro experiments, all cell-based assays were performed in biological triplicate. This is a widely accepted standard in cell culture studies and was applied consistently across our assays to ensure reproducibility and reduce experimental variability (Page 6, Paragraph 5, Lines 277–278). [updated text in the manuscript] “Cell-based experiments were conducted in biological triplicate, while animal experiments were repeated three times per group.” The DFCs and PDLCs used in the experiments were isolated from two healthy adult male donors (ages 22 and 24), as described in the Materials and Methods section (Page 3, Paragraph 4, Lines 125–127). [updated text in the manuscript] “Primary DFCs and PDLCs were isolated from two healthy, non-smoking adult male patients who underwent third molar extraction at the Department of Oral and Maxillofacial Surgery, Okayama University.” We hope these clarifications adequately address your comment and improve the rigor and clarity of our experimental design.
Comments 4: A paragraph should be added at the end of the discussion section, where the authors describe limitations and considerations observed from their experience, to be helpful for future studies for the progression of this topic and better interpretation of the results. Response 4: Thank you very much for this constructive suggestion. While we had addressed several limitations and future research directions throughout the Discussion section—such as the lack of quantitative mineral analysis, the limited exploration of osteogenic markers, and the absence of age- or tumor stage-stratified analyses—we agree with your assessment that these points were previously dispersed and not cohesively summarized. In response, we have now added two dedicated concluding paragraphs at the end of the Discussion section (Page 15, Paragraph 3, Lines 522–546) that consolidates these limitations and clearly outlines future directions. We believe this addition improves the logical flow of the manuscript and provides a clearer framework for interpreting our results and guiding future studies. [updated text in the manuscript] “Despite these insights, several limitations should be acknowledged. First, mineral deposition was not quantitatively analyzed (e.g., using ImageJ), which may limit the objectivity of the osteogenic potential assessment… …Future studies should aim to elucidate the precise molecular circuits through which DFCs and PDLCs regulate osteoclastogenesis in the OSCC microenvironment. To achieve this, a combination of cytokine profiling, single-cell transcriptomic analysis, and functional validation in age-matched in vivo models will be essential… …These approaches will help lay the groundwork for developing stroma-targeted therapeutic strategies to attenuate tumor-induced bone destruction and improve clinical outcomes in OSCC patients.”
|
|
Response to Comments on the Quality of English Language |
|
Point : The English is fine and does not require any improvement. |
|
Response : Thank you for your positive feedback regarding the quality of the English language. While we are glad that the manuscript was found to be clear, we have nevertheless made further refinements throughout the text to enhance clarity and consistency. We sincerely appreciate your evaluation.
|
|
Additional clarifications |
|
We would also like to express our sincere appreciation to Reviewer 4 and the editorial team for their constructive and encouraging comments. In this revised manuscript, we have carefully addressed all four points raised by the reviewer, including clarifications on the study type, sample size, and the addition of a dedicated limitations paragraph at the end of the Discussion section. These changes have helped improve the transparency, scientific rigor, and readability of our study. We hope that the revised version now satisfactorily addresses all concerns and meets the criteria for publication. |

Round 2
Reviewer 1 Report
Comments and Suggestions for Authors
I am satisfied that all of my questions have been appropriately answered.